# Genomic and Immunologic Correlates in Prostate Cancer with High Expression of KLK2

**DOI:** 10.3390/ijms25042222

**Published:** 2024-02-13

**Authors:** Lucía Paniagua-Herranz, Irene Moreno, Cristina Nieto-Jiménez, Esther Garcia-Lorenzo, Cristina Díaz-Tejeiro, Adrián Sanvicente, Bernard Doger, Manuel Pedregal, Jorge Ramón, Jorge Bartolomé, Arancha Manzano, Balázs Gyorffy, Álvaro Gutierrez-Uzquiza, Pedro Pérez Segura, Emiliano Calvo, Víctor Moreno, Alberto Ocana

**Affiliations:** 1Experimental Therapeutics Unit, Oncology Department, Hospital Clínico San Carlos (HCSC), Instituto de Investigación Sanitaria San Carlos (IdISSC), 28040 Madrid, Spainadrian.sanvicente@salud.madrid.org (A.S.); pedro.perez@salud.madrid.org (P.P.S.); 2START MadridCentro Integral Oncológico Clara Campal, 28050 Madrid, Spainjorge.ramon@startmadrid.com (J.R.);; 3START Madrid-FJD, Hospital Fundación Jiménez Díaz, 28040 Madrid, Spainvictor.moreno@startmadrid.com (V.M.); 4Department of Bioinformatics, Semmelweis University, Tűzoltó u. 7-9, H-1094 Budapest, Hungary; 5Cancer Biomarker Research Group, HUN-REN Research Centre for Natural Sciences, Magyar Tudosok Korutja 2, H-1117 Budapest, Hungary; 6Department of Biophysics, Medical School, University of Pecs, H-7624 Pecs, Hungary; 7Departamento Bioquímica, Universidad Complutense de Madrid, 28040 Madrid, Spain; 8Health Research Institute, Ospital Clínico San Carlos (IdISSC), 28040 Madrid, Spain; 9Centro de Investigación Biomédica en Red en Oncología (CIBERONC), 28029 Madrid, Spain

**Keywords:** KLK2, prostate cancer, surfaceome, immunologic profile, T cell engagers

## Abstract

The identification of surfaceome proteins is a main goal in cancer research to design antibody-based therapeutic strategies. T cell engagers based on KLK2, a kallikrein specifically expressed in prostate cancer (PRAD), are currently in early clinical development. Using genomic information from different sources, we evaluated the immune microenvironment and genomic profile of prostate tumors with high expression of KLK2. KLK2 was specifically expressed in PRAD but it was not significant associated with Gleason score. Additionally, KLK2 expression did not associate with the presence of any immune cell population and T cell activating markers. A mild correlation between the high expression of KLK2 and the deletion of TMPRSS2 was identified. KLK2 expression associated with high levels of surface proteins linked with a detrimental response to immune checkpoint inhibitors (ICIs) including CHRNA2, FAM174B, OR51E2, TSPAN1, PTPRN2, and the non-surface protein TRPM4. However, no association of these genes with an outcome in PRAD was observed. Finally, the expression of these genes in PRAD did not associate with an outcome in PRAD and any immune populations. We describe the immunologic microenvironment on PRAD tumors with a high expression of KLK2, including a gene signature linked with an inert immune microenvironment, that predicts the response to ICIs in other tumor types. Strategies targeting KLK2 with T cell engagers or antibody–drug conjugates will define whether T cell mobilization or antigen release and stimulation of immune cell death are sufficient effects to induce clinical activity.

## 1. Introduction

Among the different therapeutic strategies in cancer that have reached the clinical setting, the development of antibodies against surface proteins of cancer cells has become one of the most successful approaches [1]. Several examples of antibodies targeting oncogenic membrane proteins have demonstrated clinical activity, opening the door to exploit the development of antibodies in different formats [2].

Antibody–drug conjugates (ADCs) are a type of complex compound in which an antibody is bound with a cytotoxic payload through a chemical linker guiding a potent agent into the tumoral cell [3,4]. This family of agents has demonstrated activity in patients in different clinical scenarios, being, at this moment, one of the most promising therapeutic strategies [4]. Much more recently, bispecific antibodies bind two different antigens: one that can be in the tumoral cell and the other that can be an immune effector receptor like CD3 [5]. This family of bispecific antibodies are termed T cell engagers due to their capacity to activate T cells and guide them against tumoral cells [5].

For all of these potential therapeutic options, the identification of tumor-associated antigens (TAAs) that are specifically expressed in the tumoral cell is a mandatory requirement. Therefore, a great effort should be made to identify specific TAAs in certain tumor types [6,7]. KLK2 is a member of the kallikreins family, which are a subgroup of serine proteases primarily expressed in prostatic tissue and are responsible for cleaving a pro-prostate-specific antigen into its enzymatically active form [8].

T cell engagers based on KLK2 are currently in early clinical development in prostate cancer [9]. In the case of prostate tumors with high KLK2 expression, it is unknown how the microenvironment can influence the immune response. In a similar manner, the genomic profile of this tumor type has not been explored in detail, in order to identify additional vulnerabilities for co-targeting or for a better patient selection [10].

In the current work, by using genomic information we evaluated the immune microenvironment and genomic profile of prostate tumors with a high expression of KLK2.

## 2. Results

### 2.1. Expression of KLK2 in Cancer

We first studied the expression of KLK2 across a wide number of solid tumors and hematologic malignancies using publicly available human genomic datasets, as described in the materials and methods section. KLK2 was specifically expressed in prostate adenocarcinoma (PRAD) compared with other solid and hematologic malignancies (Figure 1A). In fact, KLK2 was highly expressed in PRAD compared with non-transformed tissue (5956.5 and 4175.26 transcripts per million (TPM), respectively), which comprised a >1.4-fold change, as displayed in Figure 1B. Once the selective expression of KLK2 in PRAD was confirmed, we aimed to evaluate their role in cell survival. Using public genomic information, we observed a mild correlation between KLK2 expression levels and the viability of VCAP (RRID:CVCL_2235) and LNCAP-CLONE-FGC (RRID:CVCL_1379), two prostate cancer cell lines (Figure 1C). To further explore the role of KLK2 in PRAD, we analyzed KLK2 gene expression using the TCGA (The Cancer Genome Atlas) repository through a resource termed UALCAN. We explored the presence of KLK2 in relation to patient ethnic origin, given the different genomic background described between racial groups [11,12]. As can be seen in Appendix A, no significant differences were observed between the Caucasian and African American populations, although the number of patients in this last group was very small. We did not observe a significant association between KLK2 expression and Gleason score (Appendix A). Additionally, KLK2 was more expressed in those tumors that had the TMPRRS2-ERG fusion gene (Appendix A) as well as in tumors without metastases in lymph nodes (Appendix A). Of note, the expression of KLK2 in prostate tumors does not associate with a detrimental prognosis (HR = 0.96, *p* = 0.95).

### 2.2. Expression of Immune Populations in PRAD with a High Expression of KLK2

We next studied the association of KLK2 expression with the immune infiltrates in PRAD. KLK2 expression did not associate with the presence of any innate immune cell populations including neutrophils (Rho = −0.14, *p* = 0.004), dendritic cells (Rho = −0.153, *p* = 0.001), and macrophages (Rho = 0.075, *p* = 0.128) (Figure 2A). Likewise, KLK2 did not correlate with adaptive immune cell populations like CD4+ T cells (Rho = −0.244, *p* = 4.56 × 10^−07^) and B cells (Rho = 0.048, *p* = 0.332). Nevertheless, a weak correlation was seen between KLK2 and CD8+ T cells (Rho = 0.34, *p* = 1.03 × 10^−12^). In line with this finding, we studied the association of KLK2 with biomarkers of T cell activation including CD8A, CD8B, GZMA, GZMB, or PRF1 (Figure 2B). No clear association was observed for any of them (Figure 2C). An additional battery of more than 40 additional genes linked with T cell cytotoxic or dysfunctional activity was explored, and no association was identified (Figure 2C).

### 2.3. Genomic Alterations in PRAD and Expression of KLK2

We studied genomic alterations: mutations, amplifications, and deletions that occurred in PRAD with KLK2 expression (Figure 3A). A mild correlation between a high expression of KLK2 and the deletion of TMPRSS2 (Rho = 0.47, *p* = 1.34 × 10^−28^) was observed, with no other association identified, as can be seen in Figure 3A,B.

### 2.4. Transcriptomic Profile in KLK2 PRAD

We explored the transcriptomic profile in PRAD with a high expression of KLK2. We selected those genes with a fold change ≥ 1.4, and then classified the list into those that code for surface proteins and non-surface proteins.

As can been seen in Figure 4A, KLK2 highly correlated (Rho ≥ 0.5) with surface proteins like FAM174B, TSPAN1, and ANO7, and moderately correlated (Rho ≥ 0.4) with PTPRN2, GABRG3, CHRM1, LPAR3, CHRNA2, OR51E2, SLC23A1, and SLC13A3. When we studied the correlation between KLK2 and non-surface proteins (Figure 4D), we observed a positive correlation (Rho ≥ 0.5) with CREB3L4, GREB1, RDH11, and TMEM220, followed by GLB1L3, KCNH6, and TRPM4 (Rho ≥ 0.4).

We then analyzed the expression of surface and non-surface proteins in PRAD and non-transformed tissue. Among the surface proteins that correlated with KLK2, we observed that TSPAN1, OR51E2, FAM174B, ANO7, PTPRN2, CHRNA2, and CHRM1 were highly expressed with ≥32 transcripts per million in PRAD (Figure 4B). Highly expressed non-surface proteins included CREB3L4, TRPM4, and GREB1 (Figure 4E). Among them, we also observed that the surface proteins TSPAN1, OR51E2, FAM174B, PTPRN2, CHRNA2, CHRM1 (Figure 4C), and the non-surface proteins RDH11, CREB3L4, and TRPM4 (Figure 4F) showed a fold change ≥ 1.5 between PRAD and non-transformed tissue.

### 2.5. KLK2 Co-Upregulated Genes and the Clinical Outcome and Response to ICIs

We searched whether the co-upregulated genes predicted the clinical outcome in patients with PRAD. First of all, we evaluated if this gene set predicted biochemical relapse in PRAD. Therefore, we analyzed the expression levels of the surface and non-surface proteins in patients which had high levels of KLK2 and had developed a biochemical recurrence. As shown in Appendix A, the gene set was not able to predict biochemical relapse.

Then, we studied if the expression of these genes could predict the outcome in male patients treated with immune checkpoint inhibitors (ICIs). Among the surface proteins, we observed that a high expression of CHRNA2, FAM174B, OR51E2, TSPAN1, and PTPRN2 predicted a detrimental survival in the whole population of patients treated with both agents anti-PD(L1) and anti-CTLA4 (CHRNA2: HR = 1.34 Cl = 1.02–1.76, *p* = 0.033; FDR = over 50%; FAM174B: HR = 1.36 Cl = 1.05–1.76, *p* = 0.021; FDR = over 50%; OR51E2: HR = 1.39 Cl = 1.11–1.74, *p* = 0.0045; FDR = over 50%; TSPAN1: HR = 1.78 Cl = 1.41–2.24, *p* = 7.3 × 10^−7^; FDR = 1%; and PTPRN2: HR = 1.43 Cl = 1.1–1.84, *p* = 0.0066; FDR = over 50%). A similarly unfavorable prognosis was predicted via the high expression of the non-surface protein TRPM4 (TRPM4: HR = 1.88 Cl = 1.47–2.42, *p* = 4.5 × 10^−7^; FDR = 1%) (Figure 5).

Taking into consideration that each gene individually predicted a detrimental survival, we aimed to study the association of those with a clinical outcome. As shown in Figure 6A, we found that the combination of CHRNA2, FAM174B, OR51E2, TSPAN1, PTPRN2, and TRPM4 also predicted a detrimental survival in male patients treated with ICIs (HR = 1.93 Cl = 1.41–2.48, *p* = 1.3 × 10^−7^; FDR = 1%). Given the fact that these proteins were associated with a detrimental survival in response to ICIs, we explored the association of them with immune infiltrates in PRAD. No correlation was found with these genes and immune cells (Figure 6B), suggesting their role favors an inert immune microenvironment.

Finally, although this set of genes was evaluated in a global population of male patients treated with ICIs, we aimed to study their presence in prostate cancer based on the nodal status (N0 vs. N1) and presence of the TMPRRS2-ERG fusion gene. These data are presented in Appendix A. Regarding the nodal metastasis status (Appendix A), the expression of these genes was more present in those tumors with no regional lymph node metastasis than in those with nodal metastases (1–3 regional lymph nodes). And lastly, FAM174B, OR51E2, TSPAN1, PTPRN2, and TPRM4 were more expressed in those tumors that had the TMPRRS2-ERG fusion gene, just like KLK2 (Appendix A)

## 3. Discussion

In the present article, we evaluate the transcriptomic and immunologic profile of prostate cancer patients with a high expression of KLK2. The main goal of this project was to identify transcriptomic, genomic, and immunologic cell correlates present in prostate cancer with a high expression of KLK2. One of our first findings was the observation of the exclusive expression of KLK2 in PRAD compared with other tumors and its high presence in relation with normal tissue. Of note, only for two cellular models the inhibition of KLK2 impacted on cell proliferation, suggesting its limited role in cell survival. In line with this, we did not observe any association between KLK2 expression and Gleason score or tumor grade, or the presence of lymph nodes (N1). Indeed, KLK2 expression did not predict the outcome. Similarly, no differences were observed for ethnic origin.

These data clearly show the relevance of KLK2 and its exclusive presence in PRAD, positioning this protein as an excellent therapeutic target for the development of antibody-based therapies. In line with this, T cell engagers against this protein are currently ongoing, in a similar manner as antibody or radiolabeled–drug conjugates (ADCs) [9,12,13,14]. In addition, this shows that KLK2 cannot be considered as an oncogene by itself. 

To identify additional associated biomarkers of potential druggable vulnerabilities, we mapped the genomic profile associated with a high presence of KLK2. We observed only a mild correlation between a high expression of KLK2 and TMPRSS2 deletions (Rho = 0.47, *p* = 1.34 × 10^−28^), with no other association observed. Of note, TMPRSS2 deletions were exclusively present in PRAD [15,16]. TMPRSS2 is highly expressed in prostate cancer and contains androgen response elements in the promoter [17]. It must be considered that the commonly deleted portion between ERG and TMPRSS2 could contain important tumor suppressor genes [18].

When evaluating the immune profile of PRAD with a high expression of KLK2, we observed that the presence of this protein was linked with an inert immune microenvironment. There was a clear absence of immunologic cells including CD8+, CD4+ T cells, B cells, neutrophils, dendritic cells, or macrophages. In a similar manner, there was a lack of markers of T cell activation including CD8A, CD8B, GZMA, GZMB, or PRF1, among others. These data suggest that KLK2, among other proteins highly present in prostate cancer, would have a negative effect on the immune system [19,20].

We next explored the transcriptomic profile that codes for the surface and non-surface proteins highly present in tumors with a high presence of KLK2. Of note, these highly differentiated proteins did not correlate with a more aggressive phenotype, nor predict a biochemical relapse or detrimental outcome. However, taking into consideration the immunosuppressive role of these proteins, we explore the relationship between the presence of these proteins and their response to ICIs. Given the fact that no data are available in relation to the ICI’s effect in PRAD, we evaluated their role in a wide population of ICI-treated tumors, as described in the materials and methods section. We observed that a high expression of CHRNA2, FAM174B, OR51E2, TSPAN1, and PTPRN2 predicted a detrimental survival in the whole population of patients treated with both agents anti-PD(L1) and anti-CTLA4. Similar findings were observed for the non-surface protein TRPM4. Next, we observed in PRAD tumors that the expression of these proteins correlated with an inert immune microenvironment, with a lack of immune cells and markers of T cell activation. Some proteins associated with KLK2 expression induce an immunosuppressive environment leading to a lack of activity of ICIs and reinforce the concept that PRAD harbors an immune inert environment. In this context, recent data suggest the role of androgen receptor elements in the lack of activity of ICIs in melanoma [21].

Finally, we identified a clear upregulation of FAM174B, OR51E2, TSPAN1, PTPRN2, and TPRM4 in those tumors with the TMPRRS2-ERG fusion gene, including KLK2. This observation is relevant given the characteristics of this particular PRAD subtype [22]. Lastly, we observed that the PRAD with the highest presence of these genes were those without nodal involvement.

We acknowledge that our study has limitations. This is an in silico study where data were extracted from publicly available repositories. We consider that a direct evaluation using human samples will reinforce the reported findings. In addition, regarding outcome, we acknowledge that other variables could influence the patients’ outcome, including those related to the patients’ treatment and patients’ medical conditions.

Considering that KLK2 does not have an oncogenic role, its main function could be associated with the regulation of the immune system, through a potential immunosuppressive role. In this context, targeting of KLK2 with T cell engagers could enhance the presence of T cells [23]. In a similar manner, ADCs against KLK2 could inhibit the immunosuppressive function of the protein, in addition to the induction of immunologic cell death mediated by the activity of the payload [24].

## 4. Materials and Methods

### 4.1. Gene Expression and Genomic Alteration Analysis

RNA sequencing expression data to evaluate the transcriptomic levels of KLK2, TMPRSS2, and surface and non-surface proteins in tumor and normal tissue were obtained from TCGA and GTEx (Genotype-Tissue Expression) databases [25], using the bio informatics tool ‘Gene Expression Profiling Interactive Analysis’ (http://gepia2.cancer-pku.cn/#index; last accessed on 4 September 2023) [26]. Data contained at cBioportal [27,28] (http://www.cbioportal.org; last accessed on 12 July 2023) [29] were used to evaluate the pattern of gene copy number alterations (CNAs) in prostate cancer. Using UALCAN (http://ualcan.path.uab.edu/; last accessed on 18 January 2024) [30], we evaluated the mRNA level of those genes in PRAD from various angles such as patient ethnic origin, nodal metastasis status, the presence of the TMPRRS2-ERG fusion gene, and the tumor Gleason score [31].

### 4.2. CRISPR Dependency Score

The Cancer Dependency Map (DepMap) web server (https://depmap.org/portal/; last accessed 20 September 2023) [32] was used to obtain the effect of knocking out KLK2 in various prostate cancer cell lines through CRISPRCas9 technology [33,34]. Depending on the magnitude of the effect a CERES score is given. Therefore, a negative CERES score means that knocking out the gene inhibits the proliferation and survival of the cell lines. The more essential the gene, the lower the CERES score obtained.

### 4.3. Association between Tumor Immune Infiltrates and Gene Expression

TIMER 2.0 web server [35] (https://timer.cistrome.org/; last accessed on 15 July 2023) [36] was employed to investigate the co-expression of KLK2 and the genomic alterations in PRAD. Additionally, we assessed the association of KLK2 with biomarkers of T cell activation, additional genes linked with T cell cytotoxic or dysfunctional activity, and tumor-infiltrating immune cells, such as CD8+ and CD4+ T cells and neutrophils, among others. TIMER2.0 uses 6 state-of-the-art algorithms to acquire a better estimation of immune infiltration and includes 10,897 samples over 32 cancer types [37,38]. Rho and *p*-values were obtained from Spearman’s correlation test.

### 4.4. Genes Expression in PRAD with Overexpression of KLK2

Data manipulation and statistical analyses were performed using R. The prostate cancer samples from the TCGA repository were categorized into high and low KLK2 expression cohorts based on the median KLK2 expression level. The data were pre-processed, where genes with a mean expression below 100 across all samples were excluded to minimize noise. Then, the Mann–Whitney test was employed to compare gene expression levels between the high and low KLK2 expression cohorts for all remaining genes. The outcomes of the analysis are summarized in Appendix A, which provides a comprehensive overview of the genes that exhibit the significant differential expression associated with high and low KLK2 expression cohorts in prostate cancer.

### 4.5. Surfaceome

The in silico human surfaceome database [39] (https://wlab.ethz.ch/surfaceome/; last accessed on 10 September 2023) [40] was used to determine the cell location of the proteins overexpressed in PRAD with a high expression of KLK2.

### 4.6. Outcome, Prognosis Analysis, and Biochemical Recurrence

The RNA sequencing expression data of 500 patients with prostate primary tumors obtained from the TCGA-PRAD database (https://portal.gdc.cancer.gov/projects/TCGA-PRAD; last accessed on 5 September 2023) [41] was used to analyze the association of gene expression and biochemical recurrence in patients with a high expression of KLK2.

The Kaplan–Meier Plotter Online Tool [42] (https://kmplot.com/analysis/; last accessed on 10 October 2023) [43] was utilized to determine the prognostic value in solid tumors, in relation to the expression of a particular gene. We studied the relationship between different gene expression levels and the clinical outcome in patients treated with ICIs). OS was utilized as an endpoint for the outcome analysis. OS time was defined as the time from diagnosis to patient death or last follow-up. The KM plots are represented with *p*, HR, and FDR. Genes with an HR < 1 and *p* < 0.05 predict a favorable clinical outcome.

The UALCAN resource (http://ualcan.path.uab.edu/; last accessed on 21 November 2023) [30] was used to profile KLK2 expression in relation to the tumor Gleason score [31].

### 4.7. Protein–Protein Interaction

The STRING database [44] (https://string-db.org/; last accessed on 11 October 2023) [45] was used to analyze the protein–protein interaction map of KLK2.

## 5. Conclusions

In summary, we describe the transcriptomic, genomic, and immunologic profiles of PRAD tumors with a high expression of KLK2. We have observed that KLK2 PRAD tumors are immune and inert, and are associated with TMPRSS2 deletions and with other surface proteins that predict a detrimental response to ICIs. Strategies aiming to target this protein, using it to attract T cells or to be the base to guide ADCs, could enhance the immune environment through different mechanisms including T cell migration or immunologic cell death, among others. However, it is unclear if this effect alone would be enough to produce a clinical benefit.

## Figures and Tables

**Figure 1 ijms-25-02222-f001:**
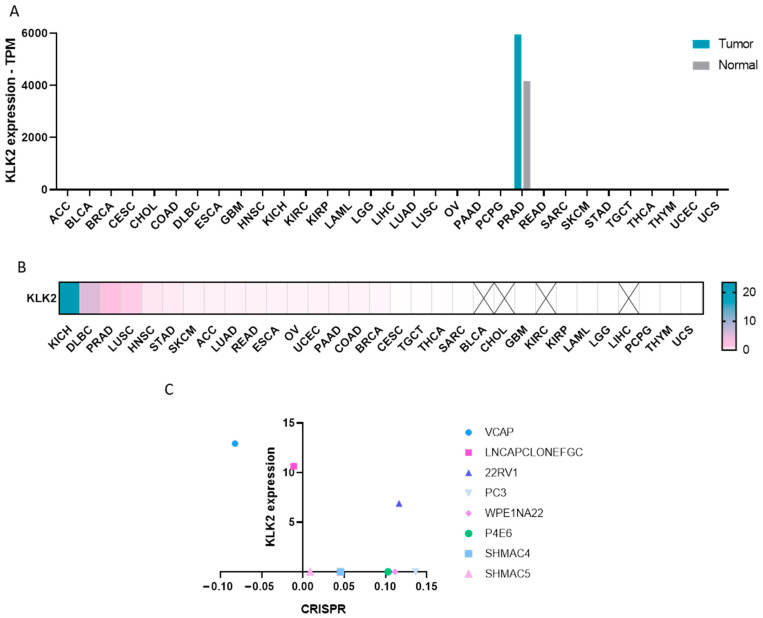
KLK2 expression profile across all tumor samples and paired normal tissues. (**A**) KLK2 expression levels in different cancers validated using GEPIA2 database. KLK2 was highly expressed (TPM > 600) in PRAD. (**B**) Heat map depicting fold change between tumor and non-transformed tissue for KLK2. The X marks the tumor types where there is no KLK2 expression. (**C**) Correlation between the KLK2 expression level and the CRISPR dependency score, using the methods described in material and methods.

**Figure 2 ijms-25-02222-f002:**
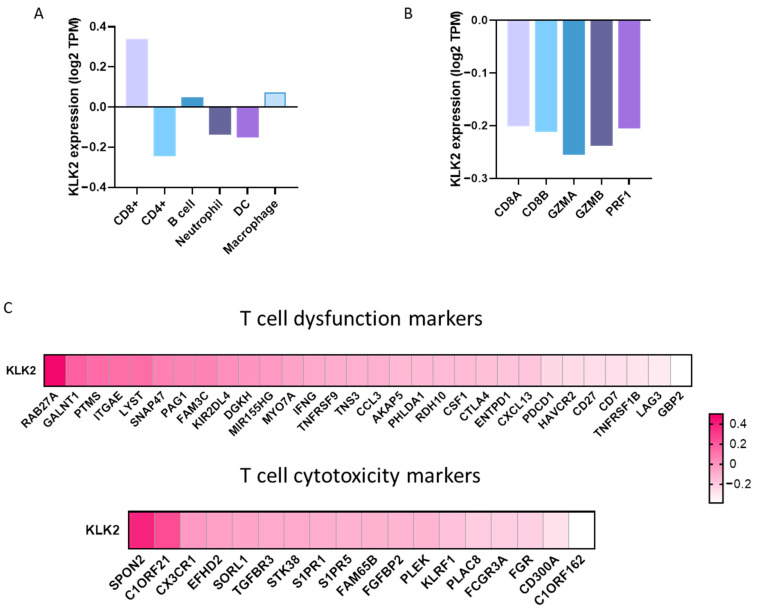
Association of KLK2 expression levels with the immune infiltrates and T cell markers in PRAD. (**A**) Correlation of KLK2 expression levels and the presence of immune infiltrates (CD8+ T cells, CD4+ T cells, B cells, neutrophils, dendritic cells (DCs), and macrophages) using the Tumor Immune Estimation Resource (TIMER2.0). (**B**) Association of KLK2 with biomarkers of T cell activation. (**C**) Heat map depicting the Spearman’s correlation coefficient between KLK2 and T cell dysfunction and cytotoxicity markers.

**Figure 3 ijms-25-02222-f003:**
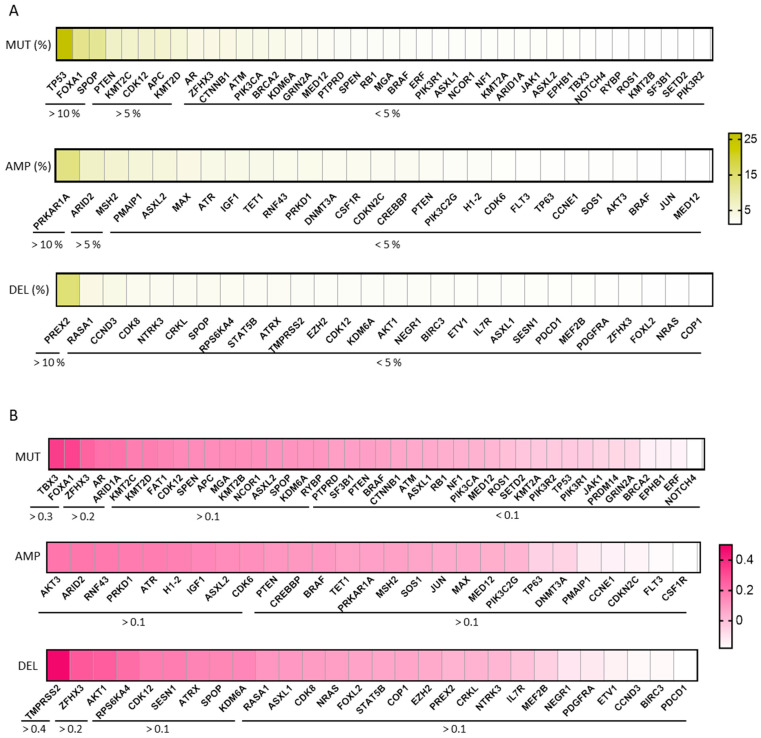
The genomic alterations more frequent in PRAD. (**A**) Heat map of the more frequent (frequency > 1%) mutations, amplifications, and deletions in PRAD using cBioPortal. (**B**) Heat map depicting the Spearman’s correlation coefficient between KLK2 and the genomic alterations more frequent in PRAD using TIMER 2.0.

**Figure 4 ijms-25-02222-f004:**
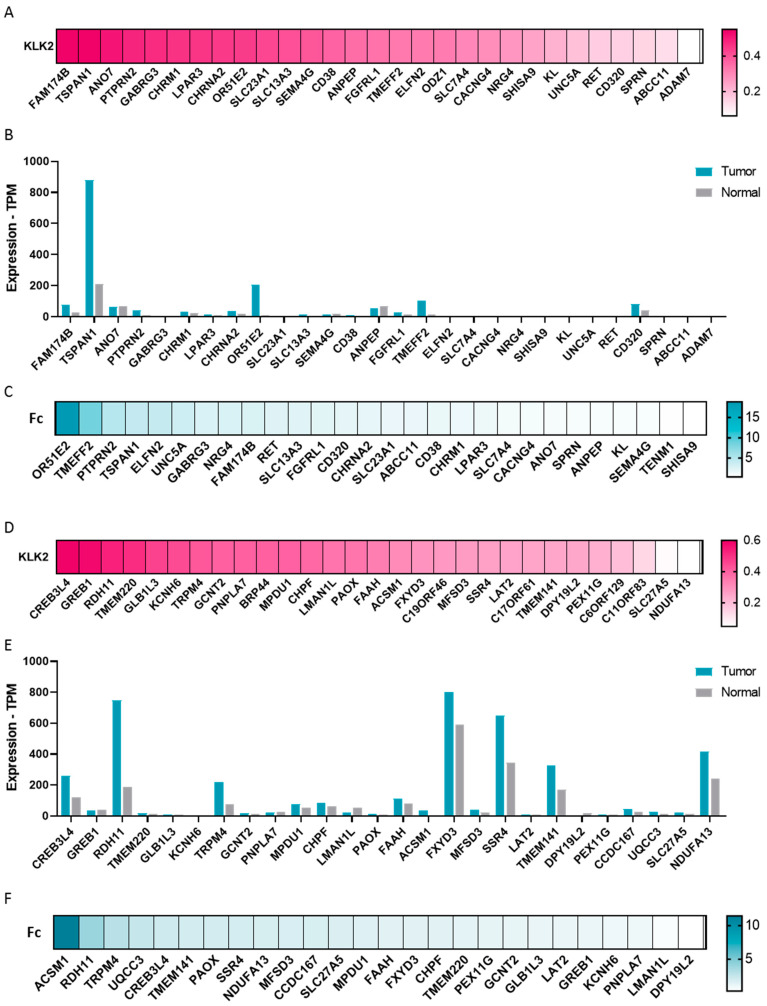
Expression profile of upregulated surface and non-surface proteins across PRAD samples and paired normal tissues when KLK2 is upregulated. Heat map depicting the Spearman’s correlation coefficient between KLK2 and (**A**) surface and (**D**) non-surface proteins in PRAD using TIMER 2.0. (**B**) Surface and (**E**) non-surface proteins expression profile across PRAD samples and paired normal tissues. Heat map depicting fold change between tumor and non-transformed tissue for (**C**) surface and (**F**) non-surface proteins.

**Figure 5 ijms-25-02222-f005:**
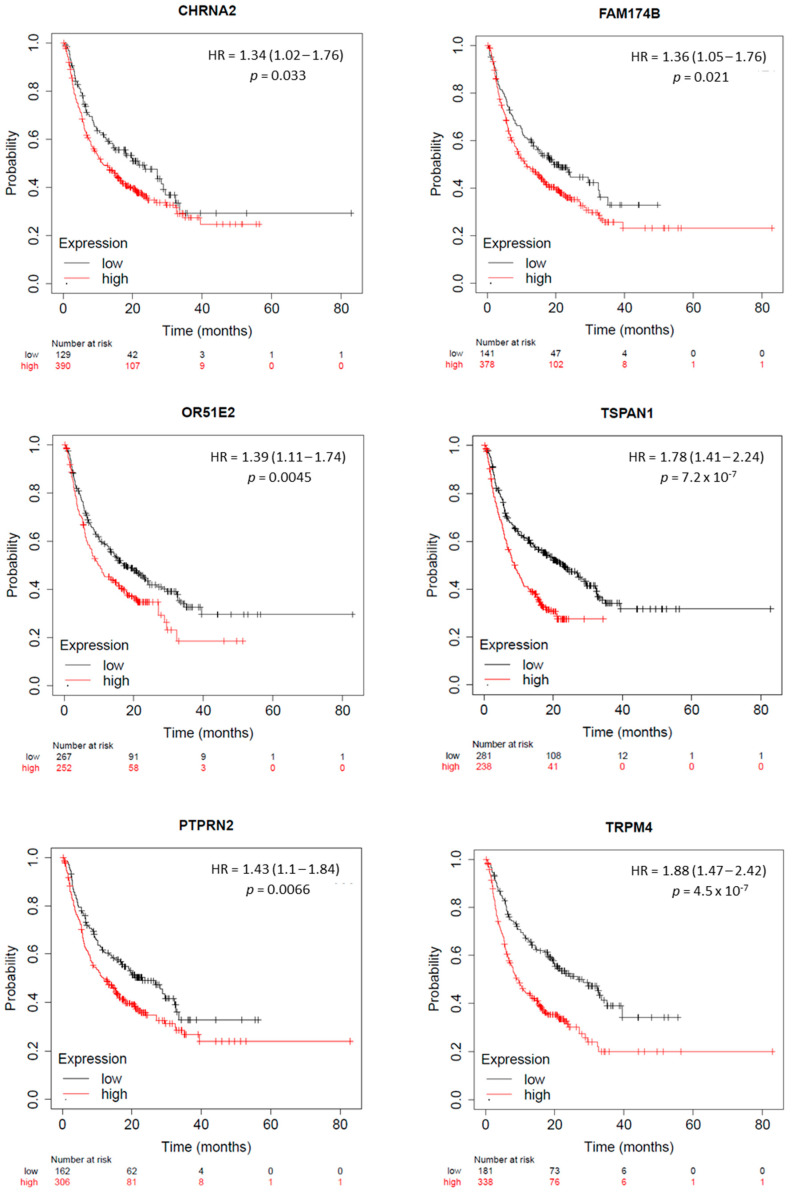
Association between surface and non-surface proteins expression levels and overall survival (OS) in male patients treated with ICIs. Kaplan–Meier survival curves comparing low and high expression of CHRNA2, FAM174B, OR51E2, TSPAN1, PTPRN2, and TRPM4. HR: hazard ratio; FDR: false-discovery rate; *p*: rank *p*-value.

**Figure 6 ijms-25-02222-f006:**
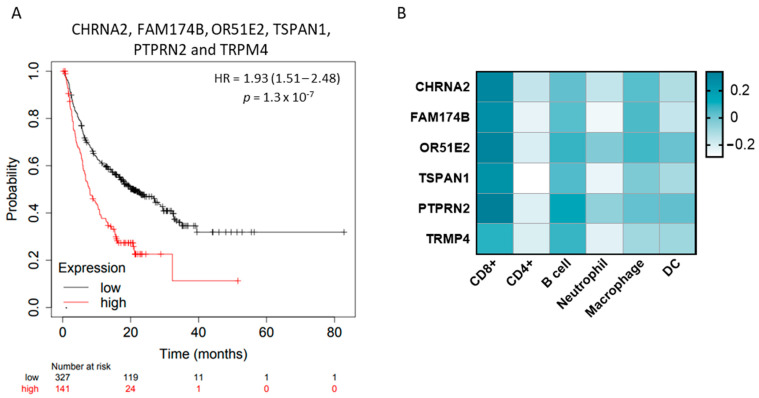
Association of surface and non-surface proteins expression levels with overall survival and immune infiltrates. (**A**) Kaplan–Meier survival curve of the association between transcriptome expression of CHRNA2, FAM174B, OR51E2, TSPAN1, PTPRN2, and TRPM4, and overall survival (OS) in male patients treated with ICIs. (**B**) Heat map depicting the Spearman’s correlation coefficient between surface and non-surface proteins expression and the presence of CD8+ T cells, CD4+ T cells, B cells, neutrophils, dendritic cells (DCs), and macrophages in PRAD using TIMER 2.0.

## Data Availability

All data generated or analyzed during this study are included in this published article.

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
