# Peer review of "Genomic and Immunologic Correlates in Prostate Cancer with High Expression of KLK2"

_ijms, 2024, doi:10.3390/ijms25042222_

Round 1
Reviewer 1 Report
Comments and Suggestions for Authors
The authors presented a rather interesting paper assessing the immune microenvironment of prostate cancer in combination with its prognostic potential based on publicly available omics data. Despite a fairly extensive study, a number of important comments can be highlighted:
1. As the main data, the authors used data from the PRAD project of the TCGA consortium, which included fairly heterogeneous data on patients, without filtering by certain criteria. This approach to analysis is incorrect, given that the molecular genetic characteristics of the occurrence and course of prostate cancer have known population differences.
DOI: 10.1038/s41416-021-01669-3
doi: 10.5732/cjc.011.10324
doi: 10.5534/wjmh.210070
2. The same remark applies to transcriptomic analysis, in relation to assessing prognosis only based on the presence of biochemical recurrence. Prostate cancer is a fairly heterogeneous disease, and the study of its molecular genetic features requires consideration in separate clinical groups. When performing the study, it is necessary to take into account such features as the presence of lymphatic metastases (N1 status), the presence of the TMPSRRS2-ERG transcript (taking into account information on the molecular genetic classification of prostate cancer in the PRAD project data and the TMPSRRS2 subtype as the most common). The transcriptomic picture when assessing biochemical recurrence will differ markedly in each case.
Based on these key observations, the authors should reconsider the main findings obtained, taking into account that the results are limited to bioinformatics analysis only.
Comments on the Quality of English LanguageThere are no serious comments on the quality of English language.
Author Response
The authors presented a rather interesting paper assessing the immune microenvironment of prostate cancer in combination with its prognostic potential based on publicly available omics data. Despite a fairly extensive study, a number of important comments can be highlighted:
- As the main data, the authors used data from the PRAD project of the TCGA consortium, which included fairly heterogeneous data on patients, without filtering by certain criteria. This approach to analysis is incorrect, given that the molecular genetic characteristics of the occurrence and course of prostate cancer have known population differences.
DOI: 10.1038/s41416-021-01669-3
doi: 10.5732/cjc.011.10324
doi: 10.5534/wjmh.210070
Response: We appreciate this comment from the reviewer. We acknowledge that there are relevant genomic differences in prostate cancer among the different ethnic populations. Therefore, we have included the mentioned references in the text. We have explored if there were differences among KLK2 expression and different ethnic groups.
We have introduced an additional supplementary figure (Supplementary figure 1A, figure S1A) in which we have explored the expression of KLK2 in Caucasian and African-american populations. No differences between groups were observed. Unfortunately the number of patients in this last population was small.
- The same remark applies to transcriptomic analysis, in relation to assessing prognosis only based on the presence of biochemical recurrence. Prostate cancer is a fairly heterogeneous disease, and the study of its molecular genetic features requires consideration in separate clinical groups. When performing the study, it is necessary to take into account such features as the presence of lymphatic metastases (N1 status), the presence of the TMPSRRS2-ERG transcript (taking into account information on the molecular genetic classification of prostate cancer in the PRAD project data and the TMPSRRS2 subtype as the most common). The transcriptomic picture when assessing biochemical recurrence will differ markedly in each case.
Based on these key observations, the authors should reconsider the main findings obtained, taking into account that the results are limited to bioinformatics analysis only.
Response: Given the fact that the datasets used does not provide all information needed, particularly for clinical. We have evaluated the presence of the transcriptional signature comparing tumors without nodal metastases and those with nodal metastases (1-3 regional lymph nodes). In the figure supplementary S3, it can be observed that the expression of these genes is more present in tumors without nodal metastases. We have added this information in the results section and we have discussed this finding in the discussion section.
Finally, we have analyzed the transcriptional levels of the gene signature described CHRNA2, FAM174B, OR51E2, TSPAN1, PTPRN2 y TRPM4 in tumors with presence of the TMPRRS2-ERG fusion gene. As can be seen in the supplementary figure S4 these genes were more present in the TMPRRS2-ERG group. We have included this data in the results section and we have explainned this finding in the discussion section.
Unfortunately, data regarding outcome was limited, and although our intention was to perform this analysis, we have not been able to execute it.
Reviewer 2 Report
Comments and Suggestions for Authors
The title should be changed because even if there are some connections between the studied parameters, it is too much to say that it “define the clinical prognostic” in patients with prostate cancer. Also, the study limitations should be mentionned in the Discussions. An extensive abreviation list should be added.
Author Response
Comments and Suggestions for Authors
The title should be changed because even if there are some connections between the studied parameters, it is too much to say that it “define the clinical prognostic” in patients with prostate cancer. Also, the study limitations should be mentionned in the Discussions. An extensive abreviation list should be added.
Response: We have modified the title, as we agree that is a very categoric assessment. The title is now written in this way:” Transcriptomic and immunologic correlates in prostate tumors with high expression of KLK2 identify clinical prognostic subgroups.
We have added some additional limitations in the discussion section and finally we have included an extensive abreviation list.
Round 2
Reviewer 1 Report
Comments and Suggestions for Authors
The authors took into account comments on working with data from the TCGA consortium
Comments on the Quality of English LanguageNo serious comments were identified regarding the Quality of English Language
Author Response
We appreciate the comments from the reviewer
Reviewer 2 Report
Comments and Suggestions for Authors
After revision the manuscript accomplish the criteria for being published.
Author Response

(The authors gave the same response as above.)
